

# Improvement of deep cross-modal retrieval by generating real-valued representation

Nikita Bhatt[1] and Amit Ganatra[2]

[1] U & P U. Patel Department of Computer Engineering, Chandubhai S. Patel Institute of Technology, Charotar University of Science and Technology (CHARUSAT), Changa, India
[2] Devang Patel Institute of Advance Technology and Research, Charotar University of Science and Technology (CHARUSAT), Changa, India

## ABSTRACT

The cross-modal retrieval (CMR) has attracted much attention in the research community due to flexible and comprehensive retrieval. The core challenge in CMR is the heterogeneity gap, which is generated due to different statistical properties of multi-modal data. The most common solution to bridge the heterogeneity gap is representation learning, which generates a common sub-space. In this work, we propose a framework called "Improvement of Deep Cross-Modal Retrieval (IDCMR)", which generates real-valued representation. The IDCMR preserves both intra-modal and inter-modal similarity. The intra-modal similarity is preserved by selecting an appropriate training model for text and image modality. The inter-modal similarity is preserved by reducing modality-invariance loss. The mean average precision (mAP) is used as a performance measure in the CMR system. Extensive experiments are performed, and results show that IDCMR outperforms over state-of-the-art methods by a margin 4% and 2% relatively with mAP in the text to image and image to text retrieval tasks on MSCOCO and Xmedia dataset respectively.

## INTRODUCTION

In the era of big data, multimedia data such as text, image, audio, and video are growing at an unprecedented rate. Such Multi-Modal data has enriched people's lives and become a fundamental component to understand the real world. We access multi-modal data in various situations like education, entertainment, advertisements, social media, which are helpful to provide effective communication. Also, real-world articles use different modalities to provide comprehensive information about any concept or topic. In recent years, Image captioning and cross-modal retrieval (CMR) have become hot research directions in vision-language tasks (*Xu, Li & Zhang, 2020*; *Yanagi et al., 2020*). The difference between them is shown in Fig. 1. The image captioning system, as shown in Fig. 1A and Fig. 1B, takes an image from the MSCOCO dataset (*Lin et al., 2015b*) and retrieves the description of an image in the form of text. Here the retrieved information is provided by both modalities (e.g., the word "cat" and pixels of "cat" are closed to each other in a learning space). On the other hand, the CMR system provides flexible retrieval where the user can give any modality as the input and retrieves any other modality as the output. As shown in Fig. 1C and Fig. 1D, an image of "owl" from the XMedia dataset

Corresponding author
Nikita Bhatt,
nikitabhatt.ce@charusat.ac.in

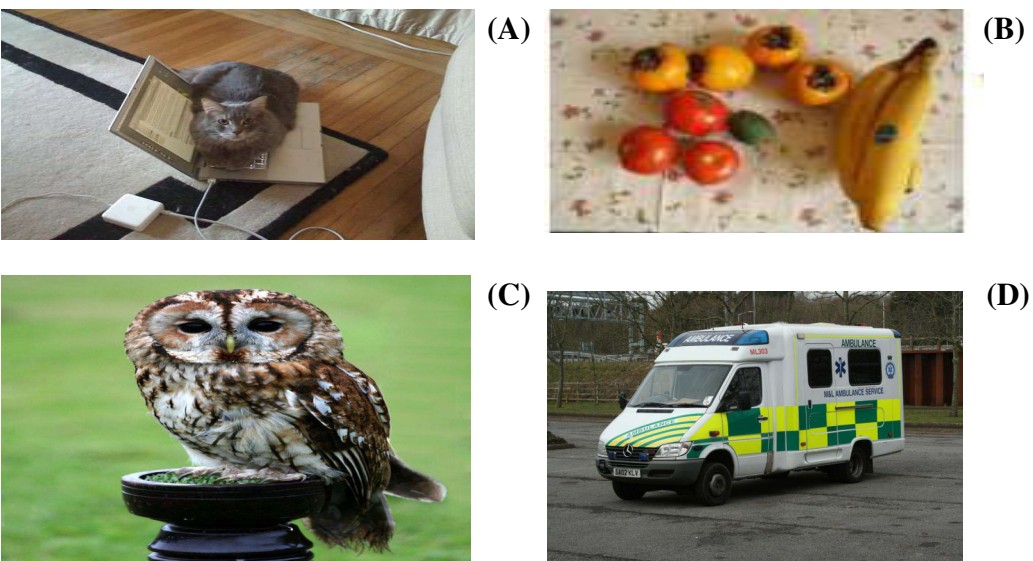

**Figure 1  Image-Text pairs from MSCOCO and Xmedia dataset.** (A) A cat is sleeping on top of an open laptop computer. (B) Tomatoes, bananas and peaches are sitting on a covered table. (C) The owl is one of the most widespread of all birds, which is found almost everywhere in the world except polar and desert regions, Asia north of the Himalayas, most of Indonesia, and some Pacific islands. (D) An ambulance is a vehicle for transportation of sick or injured people to, from or between places of treatment for an illness or injury, and in some instances will also provide out of hospital medical care to the patient.

("PKU XMediaNet Dataset", http://59.108.48.34/tiki/XMediaNet/) can be explained with multiple texts, which is not visible from the image only. Such comprehensive retrieval from the CMR system is widely used in applications like hot topic detection and personalized recommendation (*Peng et al., 2017*).

The real challenge in CMR is the heterogeneity gap (*Wang et al., 2016b*), which is generated due to the different statistical properties of each modality. For example, an image representation is real-valued and dense in the form of pixels, whereas text representation is sparse and discrete. Such a different representation of each modality does not allow a direct comparison for retrieval. The most common solution to bridge the heterogeneity gap is to generate a common sub-space (*Zhen et al., 2019*) using a function, which transforms the different representation of modalities into a common representation, such that direct retrieval is possible. Many approaches for CMR have been proposed in the past to generate a common sub-space, which is categorized into binary-valued and real-valued representation. The binary-valued representation maps heterogeneous data into the encoded form using a hash function. The advantage of binary-valued representation is less storage, which leads to faster retrieval because hamming distance can be computed faster with the help of binary code using bit operations. However, binary-valued representation suffers from information loss, which leads to unsatisfactory performance. In this paper, real-valued representation is considered, which stores actual representation. Previous CMR methods like spectral hashing (SH) (*Weiss, Torralba & Fergus, 2009*), cross-view hashing (CVH) (*Kumar & Udupa, 2011*), inter-media hashing (IMH) (*Song et al.,*

*2013*), collective matrix factorization hashing (CMFH) (*Ding, Guo & Zhou, 2014*), semantic correlation maximization (SCM) (*Zhang & Li, 2014*), Latent semantic sparse hashing (LSSH) (*Zhou, Ding & Guo, 2014*) and semantic preserving hashing (SePH) (*Lin et al., 2015a*) perform feature learning and correlation learning as an independent process to generate a common sub-space. All these CMR methods perform feature learning using scale-invariant feature transform (SIFT) (*Lowe, 2004*) and histogram of oriented gradients (HoG) (*Hardoon, Szedmak & Shawe-Taylor, 2004*). However, the correlation learning ignores the correlation between different modalities during feature learning, which may not achieve satisfactory performance. The standard statistical correlation-based method is canonical correlation analysis (CCA) (*Hardoon, Szedmak & Shawe-Taylor, 2004*), which learns linear projections from heterogeneous data, and a common sub-space is generated. However, multi-modal data is involved with non-linear relations, which cannot be learned with CCA. So, some kernel-based approach (*Bhandare et al., 2016*) has been proposed which can handle the problem, but the selection of the kernel function is one of the open challenges.

Motivated with great power and success of deep learning in the domain of representation learning, a variety of approaches have been proposed, which generates a common sub-space. The work presented in (*Ngiam et al., 2011*), proposes a deep auto-encoder (DAE) to learn the correlation between multi-modal data and a restricted Boltzmann machine (RBM) to learn a common sub-space in an unsupervised way. In (*Srivastava & Salakhutdinov, 2014*), a graphical-based model called deep Boltzmann machine (DBM) is used which does not need supervised data for training, and each layer of the Boltzmann machine adds more level of abstract information. In *Jiang & Li (2016)*, a framework called deep cross-modal hashing (DCMH) is proposed, which generates a common sub-space in a supervised way, and similarity is preserved by forcing image and text representation to be as close as possible. In *Wang et al. (2016a)*, Convolutional Neural Network (CNN) for image modality and Neural Language Model for text modality is used to learn a common sub-space using a mapping function. The Euclidean distance calculates the distance between image and text representation, which is useful for cross-modal learning. In *Zhen et al. (2019)*, a framework called deep supervised cross-modal retrieval (DSCMR) is proposed, which uses CNN for image modality and word2vec for text modality, which generates real-valued representation. A lot of work is carried out in CMR, but the performance of the CMR system can be further improved by maintaining both intra-modal and inter-modal similarity as much as possible. In this paper, we propose a novel framework called "Improvement of Deep Cross-Modal Retrieval (IDCMR)", which generates a common sub-space by preserving similarity between image and text modality. The objective function of IDCMR preserves both inter-modal and intra-modal similarity. The main contributions of IDCMR are summarized as follow:

- The proposed framework IDCMR performs feature learning and correlation learning in the same framework.
- Our proposed framework preserves intra-modal semantic similarity for text modality. Experiments are performed using various vectorization methods on

Multi-Modal datasets for the selection of an appropriate vectorization method for text modality.

- The IDCMR generates real-valued representation in the common sub-space, which preserves inter-modal and intra-modal similarities between image and text modality.
- The mean average precision (mAP) is used as a performance measure, and a comparison of the proposed framework is made with state-of-the-art methods.

The rest of the paper is divided as follows. "Background and Literature Survey" gives the background of vectorization methods for text modality. "Materials & Methods" covers the proposed model and the proposed algorithm. "Results" covers experiments and discussion. At last, "Conclusions" gives the conclusion of our work.

## Background and literature survey

The biggest challenge in natural language processing (NLP) is to design algorithms, which allows computers to understand natural language to perform different tasks. It is recommended to represent each word in form of a vector as most of the machine learning algorithms are not capable of processing text directly in its raw form. The process of converting a word into a vector is called vectorization, which represents each word into vector space. Broadly the vectorization methods are categorized into (a) local representation method and (b) distributional representation method. The most common local representation method is called bag-of-words (BoW), where each word is represented as $\mathbb{R}^{|V| \times 1}$ vector with all 0's and one 1 at the index of the word in the corpus. However, the generated matrix is sparse in nature, which is inefficient for computation, and the similarity between different words is not preserved, as the inner product between two different one-hot vectors is zero. On the other hand, in distributional representation, each word $w_i$ in the corpus is represented by featurized representation, which is denoted as $w_i \in \mathcal{R}^d$, where each word is represented in d dimensions.

The distributional representation generates distributional word vectors, which follows the concept of the distributional hypothesis (*Mikolov et al., 2013b*), which states that words that occur in the same contexts tend to have similar meanings. The distributional word vectors are generated from count-based models or prediction based models. The count-based models generate implicit distributional vectors using dimensionality-reduction techniques, which map data in the high-dimensional space to a space of fewer latent dimensions. The most popular method is singular value decomposition (SVD) (*Van Loan, 1976*), which generates embedding of each word in the vocabulary using matrix factorization, but fails when the dimensionality of matrices is very large as the computational cost for m × n matrix is O (mn²). The most popular count-based method is Glove (*Mikolov et al., 2013b*), which generates implicit vector and achieve better performance in comparison with other matrix-based methods. Another broader classification for the generation of distributional word vector is prediction based models, which are neural network based algorithms. Such models directly create low-dimensional implicit distributional representations. An example of such a model is word2vec.

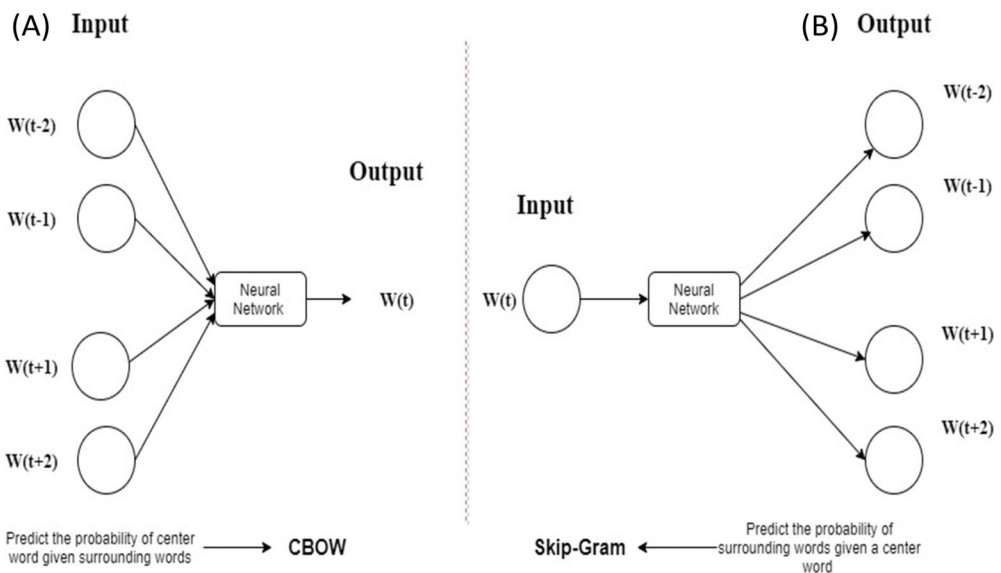

**Figure 2 Description of CBOW and SG.** (A) CBOW (B) SG.

The below section covers a detailed description of the generation of word vectors using Glove and word2vec.

## word2vec

The word2vec is a feed-forward based neural network, which has two algorithms: continuous bag-of-words (CBOW) and skip-gram (SG) (*Mikolov et al., 2013a*, *2013b*). Figure 2 (*Mikolov et al., 2013a*) shows a description of CBOW and SG where CBOW predicts the probability of center word w(t) and SG predicts the probability of surrounding words w(t + j).

## Working of SG model

SG predicts the probability of surrounding given a center word. For training of the network, there is an objective function that maximizes the probability of surrounding words given center word for each position of text t in the window size of m.

$$J'(\theta) = \prod_{t=1}^{T} \prod_{\substack{-m \leq j \leq m \\ j \neq 0}} P(w_{t+j} | w_t ; \theta) \tag{1}$$

Here $P(w_{t+j} | w_t)$ is probability of surrounding words $w_{t+j}$ given center word $w_t$. Equation (1) can be rewritten as Eq. (2)

$w_t$ can be rewritten as

$$J(\theta) = -\frac{1}{T} \sum_{t=1}^{T} \sum_{\substack{-m \leq j \leq m \\ j \neq 0}} \log P(w_{t+j} | w_t ; \theta) \tag{2}$$

$P(w_{t+j}|w_t)$ can be rewritten as $P(o|c)$, which specifies the probability of surrounding words o given center word c, and softmax function is used to generate probability.

$$P(o|c) = \frac{e^{u_0^T v_c}}{\sum_{w=1}^{V} e^{u_w^T v_c}} \tag{3}$$

where $u_0$ specifies vector representation of the surrounding word at index 0 and $v_c$ specifies the vector representation of center word. Equation (3) can be applied in Eq. (2),

$$J(\theta) = \log \frac{e^{u_0^T v_c}}{\sum_{w=1}^{V} e^{u_w^T v_c}} \tag{4}$$

Now, the objective is to optimize $v_c$ and $u_w$. So need to take the derivative with respect to $v_c$ and $u_w$.

$$J(\theta) = \frac{\partial}{\partial v_c} \log e^{u_0^T v_c} - \frac{\partial}{\partial v_c} \log \sum_{w=1}^{V} e^{u_w^T v_c} \tag{5}$$

where,

$$\frac{\partial}{\partial v_c} \log e^{u_0^T v_c} = u_0 \tag{5a}$$

$$\frac{\partial}{\partial v_c} \log \sum_{w=1}^{V} e^{u_w^T v_c} = \frac{1}{\sum_{w=1}^{V} e^{u_w^T v_c}} \times \frac{\partial}{\partial v_c} \sum_{w=1}^{V} e^{u_w^T v_c} \tag{5b}$$

$$= \frac{1}{\sum_{w=1}^{V} e^{u_w^T v_c}} \times \frac{\partial}{\partial v_c} \sum_{x=1}^{V} e^{u_x^T v_c}$$

$$= \frac{1}{\sum_{w=1}^{V} e^{u_w^T v_c}} \times \sum_{x=1}^{V} \frac{\partial}{\partial v_c} e^{u_x^T v_c}$$

$$= \frac{1}{\sum_{w=1}^{V} e^{u_w^T v_c}} \times \sum_{x=1}^{V} e^{u_x^T v_c} \frac{\partial}{\partial v_c} e^{u_x^T v_c}$$

$$= \frac{1}{\sum_{w=1}^{V} e^{u_w^T v_c}} \times \sum_{x=1}^{V} e^{u_x^T v_c} u_x$$

Combine (5.a) and (5.b),

$$J(\theta) = u_0 - \frac{1}{\sum_{w=1}^{V} e^{u_w^T v_c}} \times \sum_{x=1}^{V} e^{u_x^T v_c} u_x$$

Above equation can be rewritten as

$$J(\theta) = u_0 - \sum_{x=1}^{V} P(x|c). u_x \tag{6}$$

Where, $u_0$ is the actual ground truth and $P(x|c)$ is the probability of each surrounding word x given the center word c, and $u_x$ is the average of all possible surrounding words. So

cost function of SG guarantees that the probability of occurring surrounding words maximizes given a center word.

## Working of continuous bag-of-words model

CBOW predicts the probability of a center word given surrounding words. Input to CBOW is d dimensional one-hot vector representation of a center word. The representation of a center word is generated by multiplying d dimensional vector with the weight matrix W of size p × d where p is the featurized representation of a word.

$$h_{p\times 1} = W^T_{p\times d}x_d = V_c \tag{7}$$

The above representation is a vector representation of the center word $V_c$. The representation of outside words is generated by multiplying center representation with the weight matrix W'.

$$u_{d\times 1} = W'^T_{d\times p}h_{p\times 1} = V^T_w V_c \tag{8}$$

Where, $V_c$ is a vector representation of the center word and $V_w$ is a vector representation of surrounding words. It is a prediction-based model so need to find the probability of a word given the center word $P(w|c)$.

$$y_i = P(w|c) = \varphi(u_i) = \frac{e^{u_i}}{\sum_{i'} e^{u_i}} = \frac{e^{V^T_w V_c}}{\sum_{w'\in Text} e^{V^T_{w'} V_c}} \tag{9}$$

There is an objective function, which maximizes $P(w|c)$ by adjusting the hyper parameters i.e., $v_c$ and $v_w$.

$$l(\theta) = \sum_{w\in Text} \log P(w|c; \theta) \tag{10}$$

Put value of Eq. (9) in Eq. (10),

$$l(\theta) = \sum_{w\in Text} \log \frac{e^{V^T_w V_c}}{\sum_{w'\in Text} e^{V^T_{w'} V_c}}$$

$$= \sum_{w\in Text} \log e^{V^T_w V_c} - \sum_{w\in Text} \log \frac{1}{\sum_{w'\in Text} e^{V^T_{w'} V_c}}$$

To optimize the hyper parameter, need to take derivation with respect to $v_c$ and $v_w$.

$$\frac{\partial l}{\partial v_w} = \sum_{w\in Text} v_c - \frac{1}{\sum_{w'\in Text} e^{V^T_{w'} V_c}} \times \frac{\partial l}{\partial w} e^{V^T_{w'} V_c}$$

$$= \sum_{w\in Text} v_c - \frac{1}{\sum_{w'\in Text} e^{V^T_{w'} V_c}} \times e^{V^T_w V_c} \times v_c$$

$$= \sum_{w\in Text} v_c - P(w|c)v_c$$

$$= \sum_{w\in Text} v_c[1 - P(w|c)]$$

For optimization, gradient descent algorithm is applied and hyper parameter is optimized.

$$V_w = V_w - \eta V_c [1 - P(w|c)] \tag{11}$$

Similar steps are followed for hyperparameter $V_c$.

$$V_c = V_c - \eta V_w [1 - P(w|c)] \tag{12}$$

So CBOW and SG preserve the semantic similarity by following the distributional hypothesis in comparison with BoW model.

## Glove (Count based method)

In contrast to word2vec, Glove captures the co-occurrence of a word from the entire corpus (*Pennington, Socher & Manning, 2014*). Glove first constructs the global co-occurrence matrix $X_{ij}$, which gives information about how often words i and j appear in the entire corpus. The size of the matrix can be minimized by the factorization process, which generates a lower-dimensional matrix such that reconstruction loss is minimized. The objective of the Glove model is to learn the vectors $v_i$ (vector representation of i word) and $v_j$ (vector representation of j word), which are fruitful to information which is in the form of $X_{ij}$. The similarity between words is captured by finding the inner product $v_i^T v_j$, which gives similarity between words i and j. This similarity is proportional to $P(j|i)$ or $P(i|j)$, where $P(j|i)$ gives the probability of word j given the word i.

$$v_i^T v_j = \log P(j \,|i)$$

$$\text{where, } \log P(j \,|i) = \frac{X_{ij}}{\sum X_{ij}} = \frac{X_{ij}}{X_i}$$

$$v_i^T v_j = \log X_{ij} - \log X_i \tag{13}$$

Similarly,

$$v_j^T v_i = \log X_{ij} - \log X_j \tag{14}$$

Equations (15) and (16) are added,

$$2v_j^T v_i = 2\log X_{ij} - \log X_i - \log X_j$$

$$v_j^T v_i = \log X_{ij} - \frac{1}{2}\log X_i - \frac{1}{2}\log X_j \tag{15}$$

Here $v_i$ and $v_j$ are learnable parameters and $X_i$, $X_j$ is word specific biases, which will be learned as well. The above equation can be rewritten as

$$v_j^T v_i \,+\, b_i \,+\, b_j = \log X_{ij} \tag{16}$$

where $b_i$ is word specific bias for word i and $b_j$ is word specific bias for word j. All these parameters are learnable parameters, whereas $X_{ij}$ is the actual ground truth that can be known from the global co-occurrence matrix. Equation (16) can be formulated as an

optimization problem, which gives the difference between predicted value using model parameters and the actual value computed from the given corpus.

$$\min_{v_i, v_j, b_i, b_j} \sum_{i,j} \left( v_j^T v_i + b_i + b_j - \log X_{ij} \right)^2 \tag{17}$$

In comparison with word2vec, Glove maintains both the local and global context of a word from the entire corpus. To select an appropriate vectorization method, which maintains intra-modal semantic coherence, the below section covers experiments performed using different vectorization methods on Multi-Modal datasets. The Convolutional Neural Network (CNN) is adopted for image modality in the proposed framework, as it has shown promising performance in many computer vision applications (*Bhandare et al., 2016*).

# MATERIALS & METHODS

## Proposed framework for cross-modal retrieval

In this section, we present our proposed framework, which generates real-valued common sub-space. It also covers the learning algorithm outlined in Algorithm 1.

### Problem formulation

The proposed framework has image and text modality, which is denoted by $\Psi = \{(X_i, Y_i)\}_{i=1}^n$ where $X_i$ and $Y_i$ is image and text sample respectively. Each instance of $(X_i, Y_i)$ has a semantic label vector $Z_i = [z_{1i}, z_{2i}, \ldots, z_{Ci}] \in R^C$, where C is the number of categories. The similarity matrix $S_{ij} = 1$, if $i^{th}$ instance of image and text modality matches to the $j^{th}$ category, otherwise $S_{ij} = 0$. The feature vectors of image and text modality lie in different representation space, so direct composition is not possible for retrieval. The objective is to learn two functions, $u_i = f(x_i, \theta_x) \in \mathbb{R}^d$ and $v_i = g(y_i, \theta_y) \in \mathbb{R}^d$ for image and text modality respectively, where d is the dimension of a common sub-space. The $\theta_x$ and $\theta_y$ are hyper parameters of image and text modality, respectively. The generated common sub-space allows direct comparison for retrieval even though samples come from different statistical properties.

### Proposed framework: Improvement of deep cross-modal retrieval (IDCMR)

Figure 3 shows the proposed framework for image and text modality. The convolutional layers of Convolutional Neural Network (CNN) for image modality are pretrained on ImageNet, which generates high-level representation for each image. CNN has five convolutional layers and three fully connected layers. Detailed configuration of the convolutional layer is given in the proposed framework. Each convolutional layer contains "f: num × size × size", which specifies the number of the filter with specific size, "s" indicates stride, "pad" indicates padding, and "pool" indicates downsampling factor. The common representation for each image is generated by fully connected layers. The number in the last fully connected layer (fc8) indicates the number of neurons or dimensionality of the output layer. Similarly, the Glove model for text modality is

---

**Algorithm 1 IDCMR.**

**Input:**

$\Psi = \{(X_i, Y_i)\}_{i=1}^n$ where $X_i$ is the input image sample, and $Y_i$ is the input text sample of ith instance.

$Z_i = [z_{1i}, \ z_{2i}, \ldots, z_{Ci}] \in R^C$ where C is the number of categories $\{(X_i, Y_i) \to c_{ji}\}_{i=1}^n, \ 0 \le j \le C$

**Output:**

The image representation $U = [u_1, u_2, \ldots, u_n]$, text representation $V = [v_1, v_2, \ldots, v_n]$, hyper parameter $\boldsymbol{\theta_x}$ of image modality, hyper parameter $\boldsymbol{\theta_y}$ of text modality, a common sub-space B.

**Initialization**

$\theta_x = 0.1, \theta_y = 0.1,$batch size=128,$\eta = 0.1, \gamma = 0.1, I_x = \left\lceil \dfrac{n}{\text{batch size}} \right\rceil, I_y = \left\lceil \dfrac{n}{\text{batch size}} \right\rceil$ where n is number of training data points.

$S_{ij} = 1$ if $(Xi, Yi) \in [z_{1i}, z_{2i}, \ldots, z_{Ci}] \ 1 \le j \le C$

$S_{ij} = 0$ otherwise

**Method**

[Image Modality]

for iteration $= 1$ to $I_x$

    Step-1 Select 128 data points (batch size) from Image X and Word Vector Y

    Step-2 Calculate learned image feature $U_i = f(X_i, \theta_x)$ by forward propagation

    Step-3 Calculate the learned text feature $V_i = g(Y_i, \theta_y)$ by forward propagation

    Step-4 Calculate the discrimination loss in the label space. (Eq. (18))

    Step-5 Calculate the discrimination loss of both text and image representation in the common sub-space. (Eq. (19))

    Step-6 Calculate the modality wise invariance loss. (Eq. (23))

    Step-7 Update the linear classifier C parameters by minimizing the cost function

$C = (UU^T)^{-1} U^T S + (VV^T)^{-1} V^T S$

    Step-8 Update the parameters of image network and text network using stochastic gradient descent,

$\theta_x = \theta_x - \eta \dfrac{\partial J}{\partial \theta_x}$ and $\theta_y = \theta_y - \eta \dfrac{\partial J}{\partial \theta_y}$

end for

---

pretrained on Google News, which represents each word in form of feature vector. The text matrix is given to fully connected layers to learn the common representation for text. To learn a common representation from image and text modality, the two sub-networks share the weights of the last layers, which generate the same representation for semantic similar image and text modality. In this work, real-valued coordinated representation is generated, which preserves intra-modal and inter-modal semantic similarity. The inter-modal similarity is preserved by minimizing the (i) discrimination loss in the label space $J_1$. The prediction of label from feature spaces is possible, by connecting a linear classifier on top of each network. (ii) discrimination loss in text and image representation $J_2$, and (iii) modality-invariant loss $J_3$ in the common sub-space. Further, the intra-modal similarity is preserved by selecting an appropriate training model for each modality.

| Conv1 | Conv2 | Conv3 | Conv4 | Conv5 | fc6 | fc7 | fc8 |
|---|---|---|---|---|---|---|---|
| f:64 ×11 ×11 s:4 p:0 2 ×2 Max Pool | f:265 ×5 ×5 s:1 p:2 2 ×2 Max Pool | f:265 ×3 ×3 s:1 p:1 | f:265 ×3 ×3 s:1 p:1 | f:265 ×3 ×3 s:1 p:1 2 ×2 Max Pool | 8192 | 4096 | 4096 |

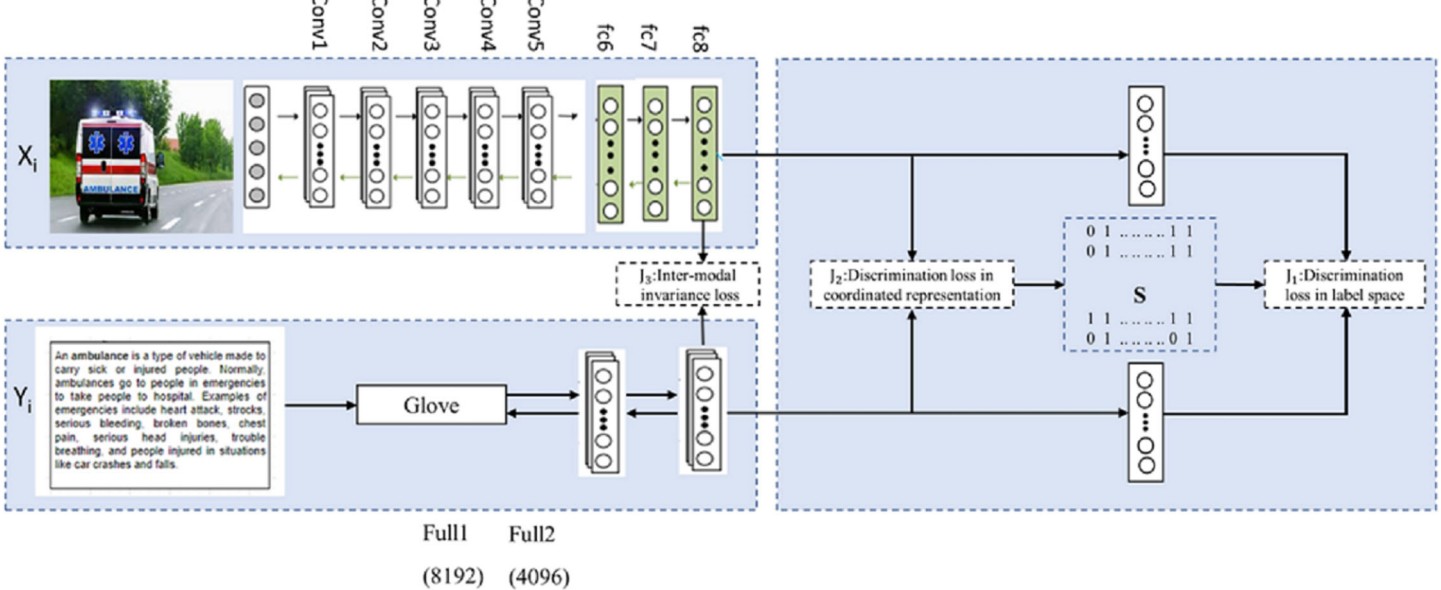

**Figure 3 Proposed framework (IDCMR).**

The biggest challenge in text modality is to preserve semantic similarities between words. There are many distributional representation methods available and the challenge is to select an appropriate method, which preserves intra-modal similarity between different words of text modality. The below section covers the learning algorithm, experiments of different distributional models, and performance comparison of the proposed framework with state-of-the-art methods.

### [a] Calculate the discrimination loss in the label space

Once the features are learned from image and text modality, a linear classifier C is connected to image and text sub networks, which predicts the semantic labels of the projected features. This predicted label should preserve the semantic similarity with label space. The discrimination loss in the label space is calculated by $J_1$ using the following equation:

$$J_1 = \frac{1}{n}\|C^T U - S\|_F + \frac{1}{n}\|C^T V - S\|_F \tag{18}$$

where, $\|\cdot\|_F$ is Frobenius norm and n is the number of instances.

## [b] Calculate the discrimination loss of both text and image modality in the common sub-space

The inter-modal similarity is further preserved by minimizing discrimination loss from image and text representation in the common sub-space, as denoted by Eq. (19).

$$J_2 = \frac{1}{n}\left[-\sum_{i,j=1}^{n}[S_{ij}\,\theta_{ij} - \log(1 + \theta_{ij})]\right] + \frac{1}{n}\left[-\sum_{i,j=1}^{n}[S_{ij}\,\varphi_{ij} - \log\left(1 + \varphi_{ij}\right)]\right] + \frac{1}{n}\left[-\sum_{i,j=1}^{n}[S_{ij}\,\varphi_{ij} - \log\left(1 + \varphi_{ij}\right)]\right] \tag{19}$$

The first part of Eq. (19), preserves the semantic similarity between image representation U and text representation V with similarity matrix S, which is denoted as,

$$\theta_{ij} = U_{*i}^T\,V_{*j}$$

The above equation should maximize the likelihood

$$P(S_{ij}\,|U_{*i}, V_{*j}) = S_{ij}\, = 1 \text{ when } S_{ij} = 1$$
$$1 - \sigma\left(\theta_{ij}\right) \text{ when } S_{ij} = 0 \tag{20}$$

where $\sigma(\theta_{ij}) = \dfrac{1}{1 + e^{-\theta_{ij}}}$ is a sigmoid function that exists between 0 to 1, and it is preferable when there is a need to predict the probability as an output. Since the probability of anything exists between a range of 0 to 1, sigmoid is the right choice.

It is represented as,

$$P(S_{ij}\,|U_{*i}, V_{*j}) = \pi\left(\sigma\left(\theta_{ij}\right)\right)^{S_{ij}}\left(1 - \sigma\left(\theta_{ij}\right)\right)^{1 - S_{ij}}$$
$$= \sum_{i,j=1}^{n}[S_{ij}\,\theta_{ij} + \log\left(1 - \theta_{ij}\right)] \tag{21}$$

Equation (21) can be rewritten as below cost function which forces representation $\theta_{ij}$ to be larger when $S_{ij} = 1$ and vice versa.

$$J = -\sum_{i,j=1}^{n}[S_{ij}\,\theta_{ij} - \log\left(1 + \theta_{ij}\right)] \tag{22}$$

So, here cost function forces $\theta_{ij}$ to be larger when and vice versa.

The second part and third part of the equation measures the similarities with image representation and text representations.

$$\varphi_{ij} = U_{*i}^T\,U_{*j}$$

$\varphi_{ij}$ is image representation, for instance, i and j whereas

$$\phi_{ij} = V_{*i}^T\,V_{*j}$$

$\phi_{ij}$ is text representation, for instance, i and j.

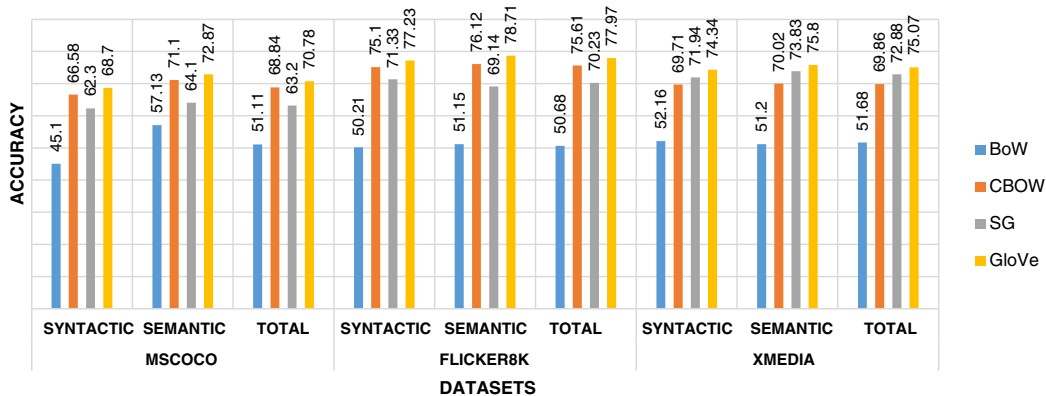

**Figure 4 Performance of vectorization methods on multi-modal datasets.**

## [c] Calculate the modality wise invariance loss

$$J_3 = \frac{1}{n}\|U - V\|_F \tag{23}$$

The final objective function is,

$$J = J_1 + \lambda J_2 + \eta J_3 \tag{24}$$

The final objective function of IDMR in Eq. (24) can be optimized during the stochastic gradient descent algorithm. The $\lambda$ and $\eta$ are hyper parameters. The $J_1$, $J_2$, and $J_3$ are the loss functions, used to preserve inter-modal similarity between image and text modality. The proposed framework has used the sigmoid activation function, which is a nonlinear function used to learn complex structures in the data. However, sometimes it suffers from vanishing gradient descent, which prevents deep networks to learn from learning effectively. The problem of vanishing gradient can be solved by using another activation function, like rectified linear activation unit (ReLU).

## RESULTS

To evaluate the effectiveness of the proposed framework, we have performed experiments on well-known datasets MSCOCO (*Lin et al., 2015b*), Flickr8k ("Flickr8K", https://kaggle.com/shadabhussain/flickr8k), and XMedia (PKU XMediaNet Dataset, Supplemental File; *Zhai, Peng & Xiao, 2014*; *Peng et al., 2016*), which are widely used in the studies. The MSCOCO dataset has total of 3,28,000 images, which is divided into 91 categories and each image is associated with at least 5 captions. The MSCOCO dataset consists of daily scene images and their descriptions. The training set consists of 15,000 images and the query set consists of 4,000 images. The Flickr8k dataset contains 8,000 images and each paired with 5 different captions. The training set consists of 6,000 images and the testing set consists of 1,000 images. The XMedia dataset has text, image, video, and audio
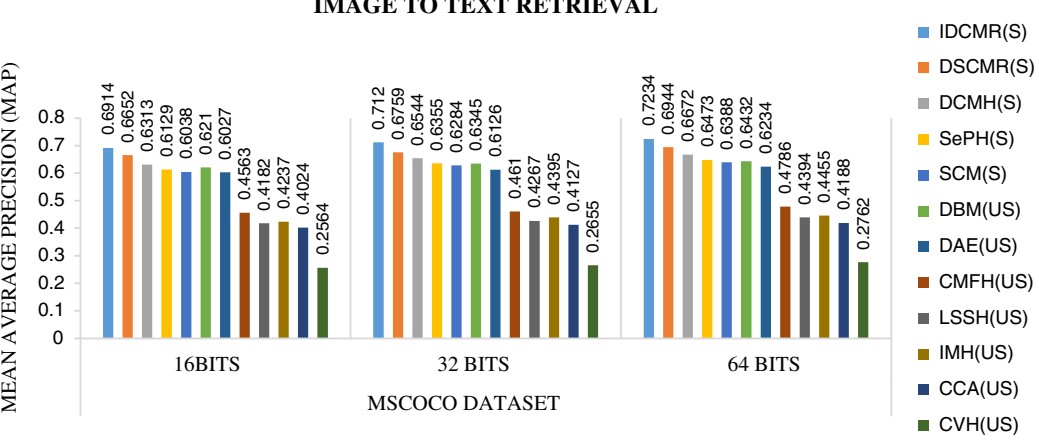

**Figure 5** Performance of CMR methods on MSCOCO dataset for image → text retrieval.

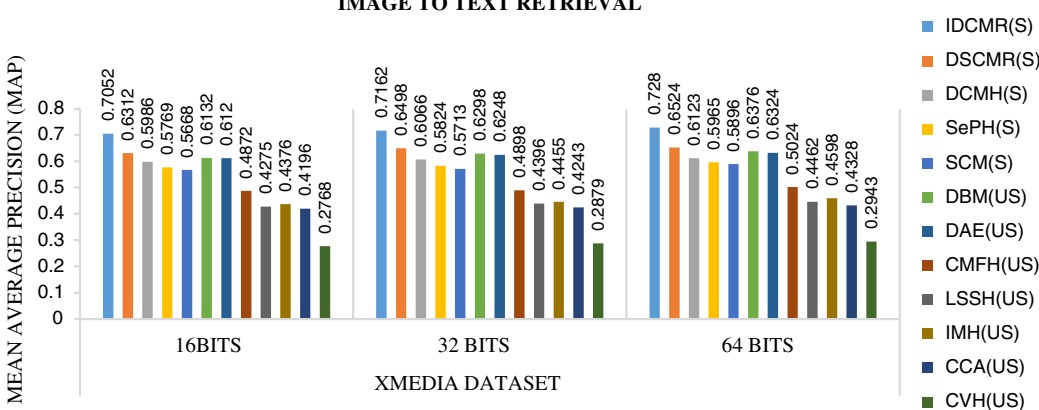

**Figure 6** Performance of CMR methods on Xmedia dataset for image → text retrieval.

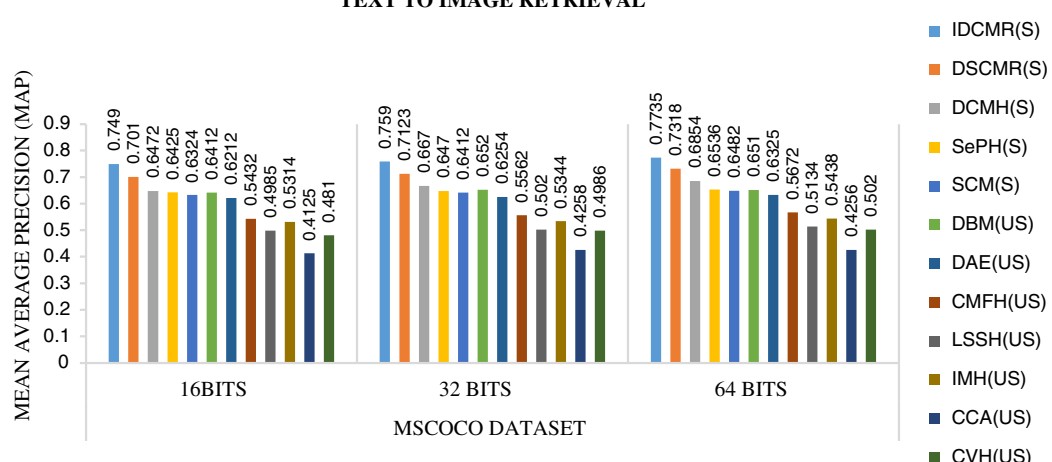

**Figure 7** Performance of CMR methods on MSCOCO dataset for text → image retrieval.

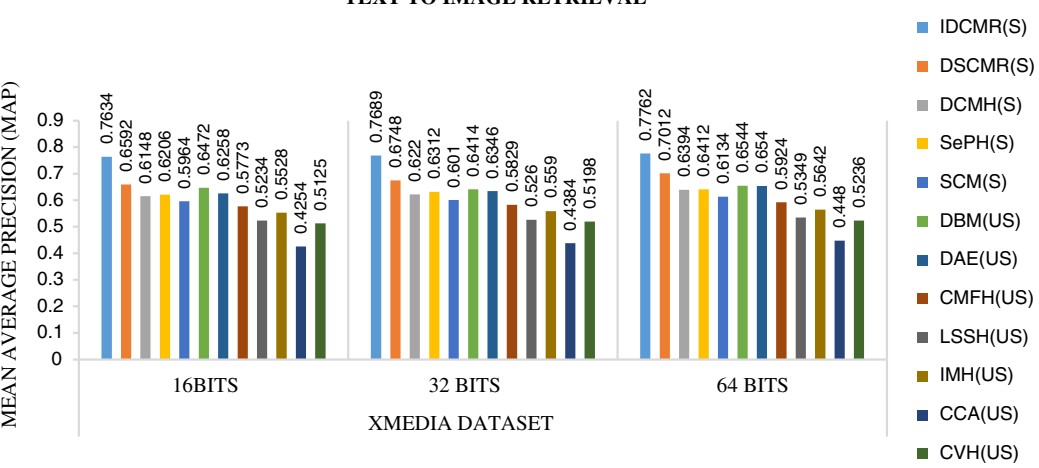

**Figure 8 Performance of CMR methods on Xmedia dataset for text → image retrieval.**

modality, which has a total of 20 different categories and each category has 600 media instances. The training set consists of 8,000 images and the testing set consists of 1,500 images. We perform experiments on GPU architecture, NVIDIA cuda cores-3840, the memory size of 12 GB GDDR5X, 32 GB RAM, 2TB hard disk, and Intel Core i7 8[th] generation. We have considered mean average precision (mAP) as a statistical measure, which is used to measure the performance of the CMR system (*Peng et al., 2017*; *Yanagi et al., 2020*).

 Following is the analysis after performing experiments.

1. The biggest challenge in text modality is to preserve semantic similarity between different words. Here experiments are carried out between different vectorization methods like BoW, CBOW, SG, and Glove. The syntactic and semantic pairs are selected from each Multi-Modal dataset like MSCOCO, Flickr8k, and XMedia. The cosine similarity is used to calculate the similarity between different pairs of words. Figure 4 shows the performance of various vectorization methods on Multi-Modal datasets. The experiment shows that Glove outperforms other vectorization methods. The Glove achieves better performance as it can preserve the similarity of words by considering the entire corpus. Due to better performance, Glove is selected as a vectorization method in the proposed framework.

2. The objective function of IDMR generates real-valued representation of image and text modality in the common sub-space, which preserves inter-modal and intra-modal similarity. The performance of IDCMR is compared with state-of-the-art CMR methods. Source codes of DCMH, DSCMR, SePH, SCM, DBM, and DAE are provided by the corresponding authors. Figures 5, 6, 7, and 8 show the performance of IDCMR on MSCOCO and XMedia dataset for image→text and text→image retrieval, respectively. The experiment shows that IDCMR outperforms over state-of-the-art methods in both image→text and text→image retrieval. The advantage of IDCMR over

other state-of-the-art methods is that the objective function of IDCMR preserves both inter-modal similarity and intra-modal similarity.

## CONCLUSIONS

The work presented in the paper has proposed a framework called "Improvement of Deep Cross-Modal Retrieval (IDCMR)", which is restricted to image and text modality. The generated heterogeneity gap is bridged by generating a common sub-space. The nature of the common sub-space is real-valued, which preserves similarities between different modalities. The uniqueness of our proposed framework is that we consider both the inter-modal and intra-modal similarities between various modalities. The proposed framework outperforms state-of-the-art methods in text→image and image→text retrieval tasks on multi-modal datasets. However, there exist many types of noise and redundancies in multi-modal data, which need to be resolved to improve the performance of the CMR system. Here the proposed framework is restricted to image and text modality, which can be extended to other modalities.

### Funding
The authors received no funding for this work.

### Competing Interests
The authors declare that they have no competing interests.

### Author Contributions
- Nikita Bhatt conceived and designed the experiments, performed the experiments, analyzed the data, performed the computation work, prepared figures and/or tables, and approved the final draft.
- Amit Ganatra conceived and designed the experiments, analyzed the data, performed the computation work, authored or reviewed drafts of the paper, and approved the final draft.

### Data Availability
Code and links for the dataset are available in the Supplemental Files.
Third-party datasets used:
Flickr8k: https://www.kaggle.com/shadabhussain/flickr8k.
MSCOCO: https://cocodataset.org/#download.
XMedia: *Zhai, Peng & Xiao (2014)*, *Peng et al. (2016)*.

### Supplemental Information
Supplemental information for this article can be found online at http://dx.doi.org/10.7717/peerj-cs.491#supplemental-information.

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
