# Peer review of "Improvement of deep cross-modal retrieval by generating real-valued representation"

_PeerJ Computer Science, doi:10.7717/peerj-cs.491_

## Round 0.1 · original submission · Minor Revisions

Dear Authors,

The paper needs improvement. Kindly make necessary corrections in the manuscript for the below reviewer comments.

1. It is not clear how the common subspace (Line 264)for image and text modalities is generated. The explanation and discussion/ mathematical analysis to that end is necessary.

2. The sigmoid activation (line 272) typically demonstrates exploding gradient behavior. How is that handled? Can any other activation be more useful instead?

3. On line 218 : ‘In this work, real-valued coordinated representation is generated, which preserves intra-modal and inter-modal semantic similarity’. Please provide a reference or solid reasoning for this.

4. The title states about real-valued representation, however, it is not so clear how it is achieved in the experimentation.

5.How is the data split carried out?

6. The uploaded code has files for DCMR, however, the results show multiple other models? Have the authors experimented and obtained the results reported in Fig 6, 7, 8? If not and these baseline results are taken directly from some standard papers, it is necessary to provide the references.

7. On lines 113-115 Authors state “ Traditionally, words are mapped into vectors using bag-of-words (BoW) but fail to preserve semantic similarity between words.”

8. Are there any disadvantages of the proposed method?

9. Conclusion is weak and I suggest the authors elaborate more to bring out the contribution and novelty of the work.

·

Basic reporting

Authors presented the paper "Improvement of deep cross-modal retrieval by generating real-valued representation".

In the paper , author have proposed the framework IDCMR that preserves both intra-modal and inter-modal similarity.

From the abstract , it's not very clear the intent and the main purpose of the paper.
It's written in the abstract that "generates real-valued representation. ", so please mention the properties, whose real-valued representation is generated by model.

Experimental design

In the Introduction section, it is mentioned that "different modalities to provide comprehensive information about any concept or topic.". So mention the different modalities here.

Figure 1 represents the modality from both the benchmark dataset or it shows other info as it's written in the introduction that "difference between them is shown in Figure 1" . So what is that difference?

In the Problem Formulation Section : "semantic similarity between different modalities are preserved in a common sub-space". What is sub-space here and explain the intent behind writing the word preserved as it's not very clear.?

What is the dimension of the similarity matrix as it seems from the problem formulation , i varies from 1...n and j varies from 1 ...C, so the size should be n x C , but then under what condition , Sij = 0 ?

Validity of the findings

No Comments

Additional comments

Good Number of models have been compared in the result section and the author has established the effectiveness of the proposed method.

From the Results and Discussion : It is not very clear why the proposed model performs well against other existing models so please explain this as well.

Elaborate the conclusion section more and clearly state 2-3 future scope of the work.

An improvement in overall improvement in quality of English writing is required at some places especially the "Introduction" and "Material and Method" section.

Reviewer 2 ·

Basic reporting

The article more or less satisfies the standards. There may be few undefined terms in the equations which may be completed. The article requires several corrections in its formatting. Many of the equations are ill formatted. Many figures should be better presented.

Experimental design

The article addresses a focused research question. It aims to generate a common representation vector that captures semantic similarity of inter-modal and intra-modal data points. Experiments are rigorously designed. Reported results a re well discussed.

Validity of the findings

The experimental validity of the findings are well established. Authors propose a deep representation techniques for images as well as text in a common subspace. Goal is to preserve semantic as well as content similarities. The method presented is a step in this direction. A neural architecture is adopted for this purpose.

Additional comments

The paper attempts to obtain a common subspace representation of two modalities namely, text and image. The common representation treats multi-modal objects as pairs and generates an unified vector representation. A neural architecture is proposed for this purpose. The architecture is an adoption of Glove and other popular neural embedding techniques.

The embedding optimise intra-modal similarity as well as inter-modal semantic proximity. A category side information is used to obtain the corresponding loss function. Stochastic gradient descent with batch normalization is used to train the network.

Results are presented for a number of benchmark cross modal retrieval data sets like Xmedia and Flickr. Standard evaluation measures are compared. The proposed method do perform comparably with state of art techniques.

In general the paper addresses and important problem and proposes an effective solution. Some of the points may be addressed by the authors.

1. Some of the recent works may be referenced. For example none of the techniques that embed relation between the entities beyond category invariance has been cited. Here is a recent reporting on this topic.
Y. Zhang, W. Zhou, M. Wang, Q. Tian and H. Li, "Deep Relation Embedding for Cross-Modal Retrieval," in IEEE Transactions on Image Processing, vol. 30, pp. 617-627, 2021

2. A number of embedding techniques like glove, word2vec has been discussed. However, there are a number of studies studying their relative efficacy. This should be taken into account while narrowing down on the embedding used for the cross modal purpose.

3. The paper formatting is extremely poor. A thorough reformatting should be done. Many of the equations are difficult to read due to formatting issues. Some of the terms are undefined.

4. Many of the figures require quality improvement too.

Reviewer 3 ·

Basic reporting

Language needs improvement.
There are some typos which needs to be removed and a thorough check for grammatical mistakes is necessary.
Check line spacing. Different for different sections.
The Figures are unclear. This may be due to conversion from one format to PDF but this needs to be dealt with.
Instead of providing the already available mathematical models of the SG, BoW etc, it will be useful to provide some analysis on these methods and the maths behind them.
The algorithm for IDCMR should be provided in algorithm format.

Experimental design

It is not clear how the common subspace (Line 264)for image and text modalities is generated. The explanation and discussion/ mathematical analysis to that end is necessary.
The sigmoid activation (line 272) typically demonstrates exploding gradient behaviour. How is that handled? Can any other activation be more useful instead?
On line 218 : ‘In this work, real-valued coordinated representation is generated, which
219 preserves intra-modal and inter-modal semantic similarity’. Please provide a reference or solid reasoning for this.
The title states about real valued representation, however, it is not so clear how it is achieved in the experimentation.

It may be good idea to provide a few sample examples of the output of the IDCMR framework.
Please specify the hyper-parameters which are used in the final model for individual dataset.
How is the data split carried out ?
It is necessary to carry out cross validation.
The uploaded code has files for DCMR, however the results show multiple other models? Have the authors experimented and obtained the results reported in Fig 6,7,8? If not and these baseline results are taken directly from some standard papers, it is necessary to provide the references.

Validity of the findings

On line 113-115 Authors state “ Traditionally, words are mapped into vectors using bag-of-words (BoW) but fail to preserve semantic similarity between words.”
Where do the recent word embeddings such as GloVe, BERT, Word2Vec fit into this?

Fig 4: are these results obtained by the authors? If not, please provide the references for these values.

Is there any other method for evaluation other than mAP computation? This methods appears inadequate and does not support the claims.

Are there any disadvantages of the proposed method?
Conclusion is weak and I suggest the authors to elaborate more to bring out the contribution and novelty of the work.

Additional comments

The paper has good potential. The paper can be improved; provided the comments are addressed by the authors.

---

## Round 0.2 · accepted · Accept

All the suggestions are incorporated in the revised article.

Reviewer 3 ·

Basic reporting

No comments

Experimental design

I am generally satisfied with the responses of the authors.

Validity of the findings

I am satisfied by the responses of the authors

Additional comments

None